# The Importance of α-Klotho in Depression and Cognitive Impairment and Its Connection to Glutamate Neurotransmission—An Up-to-Date Review

**DOI:** 10.3390/ijms242015268

**Published:** 2023-10-17

**Authors:** Patrycja Pańczyszyn-Trzewik, Ewelina Czechowska, Katarzyna Stachowicz, Magdalena Sowa-Kućma

**Affiliations:** 1Department of Human Physiology, Institute of Medical Sciences, Medical College of Rzeszow University, Kopisto 2a, 35-959 Rzeszow, Poland; ppanczyszyn@ur.edu.pl (P.P.-T.); esujkowska@ur.edu.pl (E.C.); 2Department of Neurobiology, Maj Institute of Pharmacology, Polish Academy of Sciences, Smetna 12, 31-343 Krakow, Poland; stachow@if-pan.krakow.pl; 3Centre for Innovative Research in Medical and Natural Sciences, Medical College of Rzeszow University, Warzywna Street 1A, 35-595 Rzeszow, Poland

**Keywords:** Klotho, depression, cognition, oxidative stress, Nrf2, Glu, glutamate receptors, NMDAR, animal models of depression

## Abstract

Depression is a serious neuropsychiatric disease affecting an increasing number of people worldwide. Cognitive deficits (including inattention, poor memory, and decision-making difficulties) are common in the clinical picture of depression. Cognitive impairment has been hypothesized to be one of the most important components of major depressive disorder (MDD; referred to as clinical depression), although typical cognitive symptoms are less frequent in people with depression than in people with schizophrenia or bipolar disorder (BD; sometimes referred to as manic-depressive disorder). The importance of α-Klotho in the aging process has been well-documented. Growing evidence points to the role of α-Klotho in regulating other biological functions, including responses to oxidative stress and the modulation of synaptic plasticity. It has been proven that a Klotho deficit may contribute to the development of various nervous system pathologies, such as behavioral disorders or neurodegeneration. Given the growing evidence of the role of α-Klotho in depression and cognitive impairment, it is assumed that this protein may be a molecular link between them. Here, we provide a research review of the role of α-Klotho in depression and cognitive impairment. Furthermore, we propose potential mechanisms (related to oxidative stress and glutamatergic transmission) that may be important in α-Klotho-mediated regulation of mental and cognitive function.

## 1. Introduction

Depression is recurrent, and the most common mental disease manifests as depressed mood and the loss of pleasure or interest in activities for a long time. According to the WHO, approximately 280 million people worldwide suffer from depression, i.e., 3.8% of the general population, including 5% of the adult population. Depression is the leading cause of suicide. Approximately 700,000 suicides are committed per year. Importantly, suicide is the fourth leading cause of death among people aged 15–29 [1]. This makes depression a huge burden for individuals, their families, society, and the economy. Numerous options are available for the pharmacotherapy of depression; however, their efficacy is often limited, and unwanted side effects may occur. Traditional antidepressants regulate the levels of monoamines (including serotonin and norepinephrine) in the central nervous system and require long-term continuous (usually several weeks) use to produce a clinically significant therapeutic effect. Unfortunately, two-thirds of patients are prone to relapse or do not respond to such treatment [2]. It is estimated that approximately 20% of patients with major depressive disorder (MDD) develop chronic forms of depression. Recently approved by the FDA (2019), ketamine (N-methyl-D-aspartate (NMDA) receptor antagonist with additional effects on α-Amino-3-hydroxy-5-methyl-4-isoxazolepropionic acid (AMPA) receptors, L-type voltage-dependent calcium channel (L-VDCC), hyperpolarization-activated cyclic nucleotide-gated (HCN) channels, opioid receptors, and monoaminergic receptors) is an essential drug for the treatment of drug-resistant and suicidal depression, but its use is associated with many side effects [2,3,4,5]. Nevertheless, the success of ketamine as a fast-acting antidepressant has been an important impetus for extensive research into new antidepressants, the mechanisms of action of antidepressants (especially regarding their role in synaptic targets), and the pathomechanisms of depression [2,3,6].

Depression is most often accompanied by cognitive impairment. Sometimes, the loss of higher mental functions may dominate the clinical picture and significantly impact patient functioning [7]. For many years, cognitive impairment in depression has often been downplayed. Still, recently, mainly due to its high frequency in the acute stages of depression, there has been a significant increase in interest. It is suggested that the presence of cognitive symptoms in patients with MDD may significantly determine the quality of life and the risk of relapse. Therefore, the treatment of depression should also be aimed at combating cognitive symptoms. In addition, scientific research is increasingly focusing on developing new drugs that reduce depression and cognitive symptoms [2,3,4,5,7,8]. Similarly, new potential molecular targets for this type of therapy are being identified [2,3,4,5,7,8,9,10].

Klotho is a relatively recently described transmembrane protein related to β-glucuronidase, which regulates aging [11]. Klotho-deficient mice showed various changes resembling those in patients with premature aging syndromes, such as atherosclerosis, osteoporosis, age-related skin changes, ectopic calcifications, and infertility [11]. In addition, it has been shown that low Klotho protein levels (with a concomitant increase in FGF23 levels) can also be a biomarker of kidney damage. Moreover, its role as a potential indicator of cardiovascular disease (CVD) risk [12], insulin resistance, and type I diabetes complications has recently been widely discussed [13]. The Klotho protein exhibits cardioprotection at the cellular and tissue levels by contributing to the antioxidant response of cells and the protection of cardiac contractile proteins [14]. It has been shown that Klotho deficiency can also promote several neuropathologies related to the central nervous system, including loss of synapses and modulation of their plasticity, behavioral disorders, neuritis, and neurodegeneration [15]. On the other hand, an increase in the level of Klotho protein is positively correlated with lifespan (an increase of 30.8%) and resistance to oxidative stress [16]. In addition, elevated levels of Klotho in a mouse model of Alzheimer’s disease (AD) have been associated with reduced cognitive deficits and improved cognition in old and young mice [11,17,18,19,20].

Because considerable evidence has confirmed the importance of Klotho in both depression and cognitive impairments, which often coexist, it is suggested that Klotho may be a molecular factor linking these disorders [21].

This review aimed to discuss the role of the Klotho protein in the pathogenesis of depression and cognitive impairments. Eligible studies were identified by searching web-based databases (PubMed/MEDLINE, Google Scholar, Cochrane Library) using the following search terms: Klotho; depress*; bipolar*; cognitive funct*; cognitive impair*; Nrf2 depress*; glutamate; NMDA receptor; MDD; BD. In addition, reference lists from the retrieved articles were reviewed.

We hypothesize a strong connection between Klotho with glutamatergic neurotransmission and oxidative stress. Moreover, Klotho is a promising molecular target for future pharmacotherapy of depression, especially in patients with severe cognitive impairment.

## 2. Klotho—Molecular Organization of the Gene and Its Expression

Klotho is primarily produced in the kidney (renal tubules). In addition, it is present in the brain parenchyma, reproductive organs, pancreas, blood vessels (also expressed in peripheral blood circulating cells), parathyroid glands, bladder, intestines, muscles, and inner ear [22,23,24,25]. Klotho is secreted into the serum and cerebrospinal fluid (CSF) and excreted into the urine [26,27]. In the brain, the highest Klotho expression is found in the choroid plexus and Purkinje cells of the cerebellum. At the same time, in other areas (including the cortex, hippocampus, olfactory bulb, substantia nigra, and medulla oblongata) it is lower [11,22,28,29,30]. Depending on age, Klotho expression is reduced in the brains of monkeys, rats, and mice and the cerebrospinal fluid of humans [31,32]. It was found that Klotho protein function begins in fetal life, gradually increases until adulthood, and then decreases with age, mainly affecting oligodendrocytes and myelin [22,31,33,34]. Klotho protein deficiency in mice resulted in a decrease in the number of oligodendrocytes and a weakening of myelin formation [35,36]. In contrast, overexpression of the *KL* gene resulted in increased myelin formation [37]. Klotho deficiency leads to increased oxidative stress (OS) in the hippocampus and levels of markers of programmed cell death [20,33], accompanied by impaired memory and learning processes in mice aged 6 to 7 weeks [38]. On the other hand, an increase in *KL* gene expression is associated with an improvement in cognitive processes and a reduction in OS markers [18,36].

Klotho belongs to the type I transmembrane proteins encoded by the *KL* gene located on the long arm of chromosome 13 (13q12) in humans [11,39,40,41]. Despite the different localizations of the *KL* gene in humans, mice (on chromosome 5), and rats (on chromosome 12), it shows high cross-species similarity (see [40] for review). PDS5B and STARD13 sequences flank it, and the coding region contains five exons and four introns (Figure 1). Interestingly, genes homologous to both *KL* and *PDS5B* and *STARD13* have also been identified in lower species such as *Danio rerio* and *Caenorhabditis elegans*, suggesting its primary biological function. *KL* expression is regulated by several transcription factors, such as Sp1, Oct-1, Ap-2, Mzf-1, and PAX-4, binding at different sites in the 500 bp promoter region. The human promoter of the *KL* gene is particularly rich in Sp1, which collaborates with Oct-1 (results in the downregulation of *KL* expression) but lacks the TATA box, which is typical for eukaryotic genes [40]. However, this region is rich in CpG islands and sensitive to DNA methylation, which may be crucial in regulating *KL* expression [42,43]. An important factor regulating the expression of the *KL* gene is also tumor necrosis factor-alpha (TNF-α), which inhibits its transcription by producing the nuclear factor κ-light chain enhancer of activated B-cell (NF-kB). TNF-α induces the translocation of IKB-α (nuclear factor of kappa light polypeptide gene enhancer in B-cells inhibitor *alpha*; NFkB inhibitor) from NFkB-β, which allows NFkB-β to enter the nucleus [44,45]; this mechanism will be described in more detail below. In addition, the role of other factors (e.g., vitamin D, epidermal growth factors, peroxisome proliferator-activated receptor gamma—PPARγ, erythropoietin, family A homologs of ras genes, rapamycin, statins, fosinopril, and losartan) in *KL* gene regulation has been widely discussed [46,47,48,49,50,51,52,53,54,55]. Importantly, over ten single nucleotide polymorphisms (SNPs, including G-395 A/rs1207568, F352 V/rs9536314, and C1818 T/rs564481) have been identified in the human KL gene, and their significant association with the etiology or treatment of many diseases (e.g., neurological and cancer) has been confirmed [56,57].

Human Klotho is a single-pass transmembrane protein (1012 aa) consisting of a large extracellular domain (residues 34–981), a helical transmembrane region (residues 982–1002), and a short intracellular domain (residues 1003–1012). In the extracellular domain, the protein contains two regions of glycosyl hydrolase 1: KL1 (residues 57–506) and KL2 (residues 515–953), which may have weak glycosidase activity [41,58,59,60]. The extracellular part of Klotho can be cleaved by membrane proteases (mainly ADAM10 and 17—α secretases) to form a soluble form of Klotho (130 kDa), which is released into body fluids and acts as an endocrine hormone [26,27,61]. Additional cleavage (β-cut) results in two smaller fragments, KL1 and KL2 (65 kDa each), which are unlikely to be shed into the blood. In addition to that, another form of soluble Klotho (called secreted Klotho) has been described, which consists only of the KL1 fragment generated by alternative splicing (Figure 1). However, the vast majority (if not all) of soluble Klotho is believed to come from the cleavage of membrane-bound protein (called shed Klotho). This was confirmed by a significant decrease in serum Klotho concentration after the administration of secretase inhibitors [27,62]. So far, the regulation of Klotho shedding is poorly understood. In particular, it concerns the functioning of ADAM10/17 secretases, whose expression is modified by various endogenous factors. For example, growth factors, cytokines, and insulin increase their expression levels. Conversely, tissue inhibitors of metalloproteinases block the action of these enzymes. Some drugs may also have similar effects (see [27] for review).

The best-known physiological function of alpha Klotho is the interaction with fibroblast growth factor receptors (FGFR1c, -3c, -4), significantly increasing their affinity to FGF23 (fibroblast growth factor 23). This indicates that both membrane-bound and soluble Klotho forms may act as co-receptors for FGF23 [26]. Generally, FGFR activation results in the induction of PI3K/Akt, phospholipase Cγ (PLCγ), and Ras/MAPK/ERK pathways [63,64,65,66,67], but the final effect of its stimulation is isoform-dependent. For example, activated FGFR1c affects renal phosphate and calcium exchange by inhibiting sodium phosphate transporters NPT-2a and NPT-2c in the proximal renal tubules, thereby reducing inorganic phosphate reabsorption [68]. Other known biological functions of Klotho (unrelated to FGFR) include inhibition of transforming growth factor β (TGF-β) and NF-κB, activation of the nuclear factor (erythroid-derived 2)-like 2 (Nrf2) antioxidant pathway, or inhibition of insulin-like growth factor 1 (IGF-1) pathway [27].

## 3. Klotho in Depression

Research on the significance of Klotho in depression is relatively rare (especially in animal studies), which is surprising given that depression is associated with accelerated cellular aging and aging-related phenotypes and comorbidities seen in Klotho deficiency. In addition, Klotho is present in brain structures closely related to the pathophysiology of depression (including the hippocampus and cerebral cortex) and is essential in the regulation of oxidative and inflammatory processes (well-known pathological factors in depression) [15,22,27,28,29]. Moreover, higher Klotho levels are associated with enhanced prefrontal and temporal cortex network connectivity, larger brain volume, and better function [69,70].

### 3.1. The Role of Klotho in Animal Models

Depression is multifacial and conditioned by many factors, among which stress is considered one of the most important predisposing factors [71]. The first study to show that Klotho levels could be sensitive to stress factors was by Sathyanesan et al. (2012), who demonstrated its reduced expression in the brains of rats subjected to chronic unpredictable stress [72]. Similar changes were also observed in the nucleus accumbens (NAc) and the hippocampus of mice susceptible to chronic social defeat stress (CSDS—an animal model of depression assessed using a social interaction test) compared with controls or unsusceptible groups [73] (see Table 1). Genetic knockdown of the *KL* gene in the NAc resulted in depression-like changes in naïve mice. In contrast, its overexpression induced an antidepressant-like effect in normal mice and improved behavioral responses in CSDS-susceptible mice, which was associated with the modulation of the GluN2B-containing NMDA receptor. *KL* gene knockout resulted in a selective reduction in GluN2B expression comparable to that observed in CSDS-susceptible mice. In addition, overexpression of Klotho in the NAc reversed the reduction in GluN2B expression, accompanied by a change in synaptic and structural plasticity in the NAc of CSDS-susceptible mice. The use of a specific GluN2B antagonist (Ro 25-6981—a highly potent and selective blocker of NMDARs containing the NR2B subunit; bilateral infusion into the NAc at 0.1 µM, 0.5 µL) abrogated the beneficial effects of increasing Klotho level in CSDS-susceptible mice [73]. Moreover, Tan et al. (2023) noticed that *KL* gene expression in hippocampal neurons is regulated by estrogen, which translates into the resistance of female rats to 3-week chronic unpredictable mild stress. CUMS induced depression-like behavioral changes (anhedonia- and anxiolytic-like behaviors, deficits in spatial learning and memory, as assessed using the sucrose preference test, open-field test, and Morris water maze test) only in male rats (but not in females), and these changes were correlated with decreased Klotho protein levels in the hippocampus. AAV-mediated downregulation of Klotho expression in the hippocampus of rats (both sexes) decreased stress resilience, which was observed in both females and males, being particularly pronounced in females. These results confirm that Klotho is an important factor in sex differences in stress resistance. The same study also showed that Klotho is a significant factor in the estrogen-dependent increase in the number of presynaptic vesicular glutamate transporter 1 (Vglut1)-positive clusters on the dendrites of hippocampal neurons [74,75] (see Table 2). This finding confirms the relationship between Klotho and glutamate in the pathomechanism of stress-dependent diseases, including depression. 

### 3.2. Klotho, Oxidative Stress, and Inflammation in Depression

Confirmation of the hypothesis that Klotho may be related to the pathophysiology of depression comes from an increasing number of human studies that indicate changes in Klotho protein in the cerebrospinal fluid and blood. Prather et al. (2015) showed for the first time a relationship between severe chronic stress and reduced serum Klotho levels in women [76]. Similarly, decreased levels of Klotho protein are strongly associated with the severity of oxidative stress and inflammation in the prefrontal cortex of MDD patients [77]. The regulation of oxidative stress (increased resistance to oxidative stress at the cellular level) by Klotho appears to be a critical mechanism underlying anti-aging and neuroprotective processes [78,79,80]. Over the years, multiple mechanisms underlying the neuroprotective effects of Klotho, such as the regulation of TGF-β1 or the cyclic AMP signaling pathway, have been proposed [79]. Interestingly, a growing body of evidence suggests a role for cross-talk between Klotho and Nrf2 as a novel mechanism independent of the FGF23 axis [81]. The Nrf2-related pathway is the most important intracellular mediator of oxidative stress and aging [82]. Moreover, Nrf2, a redox-sensitive transcription factor, regulates cognitive function deficits common in depression [83]. Nrf2 activation seems to be significant for the neuroprotective properties of this factor, as well as for cognitive improvement [83,84]. At the same time, enhanced generation of reactive oxygen species (ROS) and dysfunction of the main antioxidant defenses, including the KEAP1-Nrf2 system, have been observed in patients with MDD and suicide victims [85,86,87,88,89]. However, the question is: *How does Klotho regulate Nrf2 activity*? Numerous studies suggested that Klotho acts as an endogenous Nrf2 activator, preventing oxidative stress and inflammatory damage in different cell types [90,91] (see also Table 3). Maltese et al. (2017) showed that soluble Klotho induces the activity of main antioxidant defense enzymes such as *heme* oxygenase-*1* (HO-1) and peroxiredoxin *1* (Prx-1) and increases Nrf2 expression and the levels of reduced glutathione (GSH) in human aortic smooth muscle cells (HASMCs) [92]. On the other hand, Wen et al. (2022) found that recombinant human Klotho pretreatment (100 ng/mL) enhanced the antioxidant response by activating the PI3K/Akt-Nrf2/HO-1 signaling pathway in human retinal pigment epithelial cells (ARPE-19) [81]. In diabetic nephropathy, Klotho overexpression reduces oxidative stress via Nrf2 activation [91]. In the context of the molecular background of depression, the results published by Zeldich et al. (2014) and Tanito et al. (2007) seem to be relevant. These studies show that the 4 h pretreatment of neurons with α-Klotho (0.4 μg/mL) induces cytoprotective pathways and increases the expression of the main antioxidant Nrf2 target genes, such as thioredoxin 1 (Trxdr-1) and peroxiredoxin *2* (Prx-2) [78,93]. Moreover, Klotho protects hippocampal neurons from glutamate-induced oxidative disturbances via the Nrf2-antioxidant-related pathway [78]. This also provides new opportunities to combine these intracellular pathways with glutamatergic transmission. The potential role of Klotho in NF-κB regulation is equally significant [94]. Immune mechanisms are associated with oxidative stress and Nrf2 dysfunction in depression [95]. Therefore, it is important to identify targets linked to this signaling cascade. It has been found that Klotho suppresses pro-inflammatory NF-κB activation and inhibits NF-κB-dependent cytokine production [96]. Moreno et al. (2011) demonstrated the potential mechanisms of this effect, associated with TNF-α function, which promotes NFκB-dependent downregulation of Klotho expression [44]. Regarding the mechanism, significantly induced IκBα degradation was reversed by Klotho treatment in H9c2 cells (a ventricular cardiomyocyte cell line). Furthermore, nuclear translocation of NF-κB (p65) was inhibited, whereas Nrf2 nuclear accumulation was enhanced by Klotho [90]. Additionally, studies on the safety and effectiveness of biofield therapy or TRI 360TM capsules (proprietary nutraceutical combination) in patients with one or more psychiatric symptoms (depression, anxiety, asthenia, sleep disorders, etc.) have shown an increase in the level of Klotho in blood serum, accompanied by a decrease in the level of pro-inflammatory cytokines and markers of oxidative stress [97,98]. These findings strongly suggested that Nrf2 activation and NF-κB modulation by Klotho may be relevant molecular targets for the development and pharmacological treatment of depression. Klotho regulates the signaling pathways involved in the pathogenesis of depression (especially in relation to oxidative stress imbalance, inflammation, and glutamate neurotransmission). Importantly, our previous studies showed that significantly lower levels of α-Klotho protein in the prefrontal cortex of patients with MDD were associated with decreased catalase (CAT) and sodium dismutase activity (SOD) [77]. An increase in IL-6 and TNF-α levels was also observed. These changes were accompanied by a trend toward decreased Nrf2 and increased NF-κB (p65) protein levels [85]. A better understanding of the mechanism underlying how Klotho-Nrf2- NF-κB induces antioxidant and anti-inflammatory effects may lead to the development of new pharmacological strategies for depression.

Gao et al., 2021 suggest that peripheral changes in Klotho protein concentration are more important for elderly MDD patients. While young MDD patients (also older patients with recurrent depressive episodes) showed no changes in plasma α-Klotho levels, a significant decrease was observed in elderly MDD patients, associated with increased disease severity and the number of T rs9315202 alleles. Moreover, young MDD patients manifested a significantly earlier age of onset, higher α-Klotho levels, and lower HAMD (the Hamilton Depression Scale) scores compared with elderly MDD patients [99]. Similar to the animal studies mentioned above, the importance of gender in the context of the role of Klotho protein in depression has also been established in human studies. For example, a large cross-sectional NHANES study (patients over 40 years of age) showed a negative correlation between serum α-Klotho concentration and the incidence of depression in middle-aged and elderly women (but not in men) [100]. Similarly, several other studies examining Klotho reported sex-related differences [101,102]. Interestingly, no relationship was found between a mother’s mental health (symptoms of depression, clinical depression, antidepressant use) before (or during) pregnancy and her children’s cord blood Klotho levels [103]. 

In patients with bipolar disorder (BD), higher serum Klotho levels have been reported than those in the MDD group [104]. Similarly, an increase in Klotho was observed in BD patients (both in remission and mania) compared with a control group. At the same time, there were no significant differences between the levels of this protein in patients in mania and remission [105] (see Table 4). 

**Table 3 ijms-24-15268-t003:** Association between Klotho with oxidative stress and inflammation. Summary of animal studies.

Species/Strain	Animal Model	Samples	Methods	Results—Western Blot	References
Male and female Klotho ^(−/−)^ mice	Klotho deficiency (backcrossed to 129 Sv inbred mice for more than nine generations to establish a 129 Sv congenic Klotho-deficient mouse line)	Liver extracts (cytosolic, nuclear fraction)	Western Blot	↓ cytoplasmic Nrf2 ↓ nuclear Nrf2	[106]
Male and female EFmKL46 mice	Klotho overexpression, EFmKL46 (transgenic allele)	Liver extracts (cytosolic, nuclear fraction)	Western Blot	↑ nuclear Nrf2	
Male and female Klotho ^(−/−)^ mice (8 months old)	Klotho deficiency (hypomorphic mutant; backcrossed to 129/SvJ mice for more than nine generations to achieve congenic background)	Heart (cardiomyocytes)	DHE staining Western Blot	↑ ROS ↓ nuclear Nrf2	[107]
Male C56BL/6 mice (6–8 weeks old)	Diabetic mice treated with Klotho protein (0.01 mg/kg i.p. every 48 h)	Heart	Western Blot RT-qPCR	↑ Nrf2; ↓ nuclear NF-κB p65 ↑ mRNA HO-1, NQO-1, GCLC ↓ mRNA TNF-α, Cox-2, IL-6	[90]
Male and female E18 Sprague-Dawley rats	Four-hour pretreatment with Klotho (0.4 μg/mL)	Primary hippocampal neurons	Western Blot	↑ Prx-2 ↑ Trxrd-1	[78]
Male and female Klotho ^(−/−)^ mice	Klotho deficiency (hypomorphic mutant, 129 Sv genetic background)	Lung	ELISA copper-reducing equivalents	↑ 8-OHdG ↑ total antioxidant capacity	[108]

Abbreviations: DHE staining—dihydroethidium (5-ethyl-5,6-dihydro-6-phenyl-3,8-diaminophenanthridine, hydroethidine).

**Table 4 ijms-24-15268-t004:** Association between Klotho and depression. Summary of human studies.

Group	Samples	Methods	Results	References
Chronically high-stress maternal caregivers for a child with autism spectrum disorder (n = 90) Low-stress control mothers of a typically developing child (n = 88)	Blood serum	ELISA	↓ α-Klotho (age- and severity of depressive symptoms-dependent) in highly stressed women	[76]
MDD, male (n = 9) Psychiatrically normal control, male (n = 9)	Postmortem brain tissues (prefrontal cortex; BA10)	ELISA	↓ α-Klotho accompanied by ↓ CAT, SOD, and ↑ IL-6, TNF-α levels in MDD	[77]
Male and female 20–45-year-olds with psychological symptoms (e.g., sleep disturbances, anxiety/depression/posttraumatic stress disorder, stress and confusion, etc.) (total n = 84, including n = 42 placebo and n = 42 TRI 360TM (proprietary nutraceutical combination) treatment)	Blood serum	ELISA	↑ Klotho accompanied by ↑ vitamin C and D3 metabolites, neurotransmitters (dopamine, norepinephrine, acetylocholine), and oxytocin as well as ↓ IL-1β, IL-8, TNF-α, malondialdehyde, and oxidized-LDL levels after TRI 360TM administration (improvement in psychological state)	[97]
MDD patients (n = 114) Age-matched healthy controls (n = 112)	Blood plasma	ELISA PCR	↓ α-Klotho levels in MDD ↔ α-Klotho levels in elderly recurrent and young MDD patients (vs. control) Earlier onset age, higher plasma α-Klotho levels, and lower HAMD scores in young MDD ↑ α-Klotho levels in rs9315202 T alleles carrier regardless age or sex Negative correlation between the rs9315202 T allele and disease severity in the elderly MDD patients	[99]
Depressed patients (n = 5272) ≥ 40 years of age	Blood serum	Cross-sectional study (data collected from the National Health and Nutrition Examination Survey—NHANES from 2007 to 2016)	Negative correlation between serum α-Klotho concentration and the incidence of depression in middle-aged and elderly women	[100]
Newborns (n = 72)	Cord blood	ELISA	No correlation between Klotho levels and mother’s depression No relationship between Klotho levels and infant sex, delivery specifics including gestational age, or anthropometrics at birth	[103]
MDD patients (n = 245) BD patients (n = 59)	Blood plasma	ELISA	↑ IL-1β, TNF-α, soluble TNF receptor (sTNFR)1, IL-12, and IL-10 in MDD vs. BD ↑ IL-6, sTNFR2, IL-18, IL-33, ST2 (IL1R Like 1), and Klotho in BD vs. MDD IL-1β levels in MDD patients with melancholic features (vs. without melancholia) sTNFR1/sTNFR2 ratio predicted MDD and state and trait anxiety and negative affect	[104]
BD patients (type 1; n = 40) Control (n = 30)	Blood plasma	ELISA	↑ Klotho levels in BD patients both in remission and mania (vs. control) No difference in Klotho levels between BD patients in mania and patients in remission	[105]

Abbreviations: HAMD—Hamilton Depression Scale.

### 3.3. The Role of Klotho in Antidepressant Therapy 

Growing evidence indicates that Klotho may be related to the effectiveness of antidepressant therapy, both pharmacologically and non-pharmacologically. Although Sartorius et al. (2019) showed no effect of 4-week administrations of escitalopram, venlafaxine, and buprorion on Klotho levels, another research team found a strong association between two single-nucleotide polymorphisms (SNPs) of the *KL* gene in response to selective serotonin reuptake inhibitor (SSRI) treatment in patients (>65 years) diagnosed with late-life MDD [109,110]. Significant improvement in post-treatment depressive symptoms (evaluated using 21 items of the Hamilton Rating Scale for Depression; HRSD-21 at baseline and after 6 months) was confirmed in patients with at least one minor allele of rs1207568 and a weaker response in patients homozygous for the minor allele of rs9536314 [110]. Based on this, it can be assumed that the T allele of SNP rs1207568 is a protective factor that determines a stronger response to SSRI treatment, while the G allele of SNP rs9536314 is a risk factor that weakens such therapy. Therefore, it can be hypothesized that carriers of the double-mutant allele in rs1207568 (T/T) without the double mutation in rs9536314 (non-G/G) should respond better to SSRI treatment, whereas patients without the double mutation in rs1207568 (non-T/T) and carriers of the double-mutation allele in rs9536314 (G/G) can be considered as a group of patients who will respond worse to SSRI treatment [110,111]. The importance of the Klotho protein in antidepressant therapy in geriatric patients with severe depression was also confirmed by Hoyer et al., who observed higher levels of Klotho, but only in the cerebrospinal fluid (not in serum), which was positively correlated with the number of single electroconvulsive therapy (ECT) sessions [112]. These observations seem somewhat debatable because another study showed no significant effect of ECT therapy on CSF Klotho levels. However, numerous limitations of this study (including small sample size, antidepressants/antipsychotics/lithium used, and comorbidities—Alzheimer’s disease) should be considered [113]. CSF Klotho level was also not related to the Seizure Quality Index (SQI) or the ability to predict the risk of non-response (and non-remission) in ECT [114]. Furthermore, Sartorius et al. (2019) showed no differences in serum Klotho levels after ECT therapy [109]. Similarly, the use of transcranial direct current stimulation (tDCS, a safe, non-invasive neuromodulatory therapy) did not change the Klotho level in BD patients [115]. However, it should be remembered that peripheral serum concentrations do not always adequately reflect processes in the central nervous system. Moreover, it can be hypothesized that the effect of ECT on Klotho may be strongly determined by the sex and age of the patients [109,112,113] (see Table 5). Recently, it was postulated that Klotho may play a role in the antidepressant effects of ketamine in patients with treatment-resistant depression and suicidal thoughts. Significantly higher levels of Klotho were reported in patients on day 3 after ketamine infusion. At the same time, no relationship was found between changes in Klotho concentration and changes in depressive and suicidal symptoms. On the other hand, higher baseline levels of Klotho were related to a weaker antidepressant effect of low-dose ketamine during the post-infusion follow-up [6]. These studies also support the association between Klotho and glutamatergic transmission, which has been previously suggested in animal studies [73,74]. 

## 4. The Relationship between α-Klotho and Cognition

It should be noted that the positive effects of Klotho on cognition and synaptic plasticity have been documented in young and old animals and a model of AD [116]. It is associated with NMDA-dependent neurotransmission [116], which is also involved in depression [3,117], but the mechanism may be different [73,118]; therefore, in this subsection, we focus on Klotho’s involvement in cognitive changes in an attempt to understand the mechanism underlying its efficacy. Experiments by Leon et al. (2017) in mice showed that peripheral delivery of a fragment of the mouse α-Klotho protein (αKL-F, a truncated peptide) (10 µg/kg, i.p.) for five days before experiments in the Morris water maze and Y-maze improved learning and memory in young mice in the context of spatial learning and working memory. Similar results were obtained in aging mice (18-month-old) and α-synuclein transgenic mice [116]. In addition, heterozygous Klotho (C3H) mutant mice show impaired recognition and associative memory in a novel object test [38]. Poor memory retention occurred 24 h after training rather than 1 h after exercise, which was interpreted as impaired long-term retention of new object recognition [38]. Dubal et al. (2014), using transgenic mice overexpressing Klotho (NTG C57Bl/6 mice crossed with hemizygous KL transgenic mice (line 46) expressing mouse Klotho under the EF-1α promoter), showed an association between the KL-VS Klotho genetic variant and cognitive changes (Morris water maze and Y-maze). However, there was no correlation with the age of the mice, suggesting that Klotho may improve cognition regardless of age [17].

As for human studies, the same authors conducting a meta-analysis found a correlation between the KL-VS Klotho genetic variant and increased cognition in 718 mostly Caucasian individuals [17]. The observed association was independent of age, sex, and APOEε4 allele status [17]. In contrast, Zhu et al. (2019) found no correlation between cognitive impairment and Klotho gene polymorphisms (G-395A/rs1207568 and F352V/rs9536314) in an Asian population. However, an association was found between Klotho gene polymorphisms and urolithiasis, cardiovascular disease (G-395A/rs1207568), cancer, and longevity (F352V/rs9536314) [56]. Further studies by other authors have documented low serum Klotho levels as an early predictor of atherosclerosis (50 healthy volunteers), and higher plasma Klotho levels are associated with better cognition [119,120]. 

If the mechanisms of enhanced cognitive performance are deliberate, several scenarios should be considered. It should be noted that Klotho does not cross the blood–brain barrier (BBB) when delivered peripherally. At the same time, preclinical studies have observed improvements in cognitive performance [116]. Moreover, changes in the behavior of NMDARs (GluN2B cleavage) and the enhancement of NMDA-dependent synaptic plasticity have been observed with peripheral αKL-F treatment [116]. As suggested by the authors, this phenomenon may be related to the activation of glutamatergic (Glu) signaling and the enhancement of synaptic plasticity [116]. Indeed, these suggestions have been verified using proteomic, electrophysiological, and Western Blot analyses [116]. However, the detailed mechanism of the occurring changes that result from one to the other is unknown. Research in this area is currently ongoing. Dubal et al. (2014) suggested that elevated Klotho levels may directly or indirectly increase GluN2B activity, facilitating LTP induction. These suggestions were confirmed using ifenprodil (a GluN2B antagonist at 5 mg/kg) [17]. This dose was sufficient to block learning and contextual memory enhancement in heterozygous Klotho transgenic mice. 

Several other scenarios related to cognitive improvement and Klotho can be considered: mitochondrial mechanisms with oxidative stress, myelin metabolism, and changes in Ca^2+^ homeostasis [121]. In a mouse model of amyotrophic lateral sclerosis (superoxide dismutase SODG93A), Klotho overexpression suppressed proinflammatory cytokines and neuroinflammatory markers and reduced neuronal loss. Additionally, Klotho promotes myelin production [121]. Subsequent studies have documented that oxidative stress mechanisms play a role in Klotho-mediated cognitive processes [38]. Klotho mutant mice with impaired recognition and associative memory accumulate products of lipid peroxidation and oxidative DNA damage in the hippocampus, such as malondialdehyde (MDA), 8-hydroxy-2-deoxyguanosine, increased SOD1 and glutathione peroxidase activity [38]. In addition, Klotho mutant mice express lower mRNA and protein levels of Bcl-2 and Bcl-X_L_ in the hippocampus [38]. α-Tocopherol, a potent antioxidant (150 mg/kg), administered for five weeks increased the preference for a novel object in Klotho mutant mice and restored deficits in associative fear memory, contextual memory, and tone-dependent freezing [38]. This study found a parallel reduction in MDA-positive cells in the hippocampal CA1 region [38]. When human umbilical vein endothelial cells were preincubated with Klotho protein and then exposed to TNF-α, suppression of intracellular adhesion molecule-1 (ICAM-1), and vascular cell adhesion molecule-1 (VCAM-1) expression, NFκB activation, and IκB phosphorylation were observed [122], indicating an essential role of Klotho in modulating endothelial inflammation [122]. These mechanisms may be necessary in atherosclerosis [122].

According to Shafie et al. (2020), mechanisms associated with calorie restriction (low-calorie and low-calorie–high-protein diets) establish a vital role of Klotho in neurodegeneration. Klotho levels and its co-receptor—FGF23—were elevated in the hippocampus and prefrontal cortex of rats fed high-protein, low-calorie, and low-calorie–high-protein diets [123]. In the novel object recognition test, rats fed these diets showed increased recognition speed (a working memory parameter) and reduced anxiety in elevated plus-maze and open field tests compared with rats fed a fat diet [123]. Increased Klotho expression was associated with BDNF and c-fos levels, as indicated using a Klotho inhibitor [123].

Gao et al. (2021) proposed an interesting concept: Klotho may represent a neurobiological link between depression and dementia. According to this, depression may be a risk factor for AD. The reduction in Klotho during depression through oxidative stress and inflammatory mechanisms paves the way for the development of cognitive impairment observed in AD [21]. α-Klotho can suppress oxidative stress by reducing insulin/IGF-1 signaling, FOXO phosphorylation, and nuclear translocation [21]. In turn, FOXO promotes reactive oxygen species scavenging by binding to the SOD_2_ promoter [21]. Similarly, depressed patients have been found to have reduced gene expression levels of antioxidant enzymes such as SOD and CAT [21]. However, the most-studied phenomenon in depression is changes in Glu neurotransmission; therefore, the next section will discuss this context with Klotho.

## 5. Klotho and Glutamate—The Possible Role of Klotho in Neurotransmission

Glutamate (Glu) is a primary excitatory neurotransmitter in the nervous system. In synaptic transmission, Glu is released by the presynaptic terminal and binds to specific ionotropic (AMPA, NMDA, kainate) and metabotropic (mGlu) receptors located in the postsynaptic membrane, producing various effects in neuronal networks [124]. A growing body of evidence suggests that changes in glutamate homeostasis and receptor trafficking may be crucial for the development of depression [125,126]. Moriguchi et al. (2019) conducted a meta-analysis (based on proton magnetic resonance spectroscopy studies) that showed a decrease in glutamate levels in the medial frontal cortex of depressed patients [127]. At the same time, significantly higher peripheral blood glutamate levels were noted in subjects with major depression compared with controls [128]. The glutamate hypothesis of depression is strongly supported by the antidepressant effects of specific NMDA antagonists (such as ketamine or esketamine) [129,130]. In addition, in 2019, esketamine (SPRAVATO) nasal spray was approved by the Food and Drug Administration (FDA) for adults with treatment-resistant depression (TRD) who had undergone multiple antidepressant therapies without regression of clinical symptoms. Therefore, the role of excitatory synaptic transmission of glutamate appears to be related to both the pathophysiology and pharmacology of depression [3]. However, the neurobiological and molecular mechanisms that modulate glutamate neurotransmission in depression have not yet been fully elucidated. Recent studies have suggested that the Klotho protein regulates the glutamate system. In this section, we discuss potential interactions between the biological functions of Klotho and glutamate neurotransmission. It should be noted that these aspects represent an entirely new direction in molecular research on depression.

The first target, linked to Klotho function and glutamatergic activity, is related to excitatory amino acid transporters (EAATs). EAATs comprise a class of five transporter isoforms (EAAT1–5) that are mainly expressed in neurons (EAAT3–4) and glial cells (EAAT1–2) of the central nervous system. The primary function of EAATs is glutamate transport (and uptake) across the plasma membrane [131]. Abnormal expression of EAATs may cause dysfunction in the glutamate system, leading to depressive symptoms [132]. For example, human studies have observed reduced EAAT4 transcript expression in the striatum of people with MDD [133]. Simultaneously, the expression of glutamate transporters is reduced in animal models of depression [134,135]. Zink et al. (2010) showed significantly reduced expression of EAAT2 (rodent nomenclature GLT1) in the hippocampus and cerebral cortex of learned helpless rats. Similarly, EAAT4 expression was suppressed in the helpless animal group [136]. The main question in this context remains, “How can klotho modulate Glu transmission?” The Klotho protein can affect the amount (expression) of EAAT, especially EAAT3/EAAT4 members. Almilaji et al. (2013) published the first results supporting this thesis. In their study, voltage-clamp experiments showed that Klotho cRNA significantly increased the I-glutamate (Iglu)-induced current (as a function of glutamate concentration) of EAAT3 and EAAT4 in Xenopus oocytes. Additional confocal microscopy and chemiluminescence showed that injection of cRNA constructs encoding Klotho increased the amount of EAAT3 and EAAT4 proteins in the membranes of oocytes. Finally, researchers showed that pretreatment of Xenopus oocytes expressing EAAT3 with recombinant human β-Klotho protein at a concentration of 30 ng/mL increased Iglu-induced current [137]. Similarly, Warsi et al. (2015) noted that treatment with soluble human recombinant β-Klotho protein (30 ng/mL) increased EAAT1 and EAAT2 expression in Xenopus oocytes, Iglu. The observed effect was reversed with DSAL (D-saccharic acid 1,4-lactone monohydrate; β-glucuronidase inhibitor) [138]. These results suggest that Klotho upregulates excitatory amino acid transporters and may play a critical role in regulating neuronal excitation. A growing body of evidence suggests that the antidepressant activity of some drugs or novel compounds is related to the regulation of EAAT expression [139,140]. In addition, the glutamate transporter EAAT3 is regulated by the mammalian target of rapamycin mTOR [141]. Activation of the mTOR pathway is known to be involved in the rapid effects of antidepressants [142]. The antidepressant effects of ketamine can also modulate the mTOR signaling cascade [142,143]. Given this mechanism, the role of Klotho in regulating the efficacy of antidepressants via the glutamate system cannot be ruled out. This is the only hypothesis that must be tested in vivo. However, this idea is supported by the fact that the interaction between mGluR5/EAATs/COX-2 has been documented in the mouse brain [144]. At the same time, upregulation of Klotho expression in the mouse testis was observed [145]. Thus, it can be hypothesized that these pathways regulate each other indirectly through COX-2 (a cellular component of the immune system) and Nrf2-related antioxidant pathways. Recombinant Klotho pretreatment (0.4 μg/mL per 4 h) of rat primary hippocampal neurons and HT22 cells (a mouse hippocampal neuronal cell line) protected these cells from glutamate-induced oxidative stress. In turn, primary hippocampal neurons isolated from Klotho-overexpressing mouse embryos showed higher resistance to L-glutamate (2 mM) neurotoxicity [78]. However, further studies are required to confirm these findings.

The ability to regulate the function of AMPA and NMDA receptors is another molecular pathway through which Klotho may be involved. Key evidence (discussed in detail in Section 4) suggests that Klotho improves memory and synaptic transmission through NMDA receptors (particularly the GluN2B subunit) [17]. In the context of depression, Wu et al. (2022) showed extensive evidence (discussed in detail in Section 3) on the relationship between Klotho and NMDAR and the development of depressive-like behavior, in particular pointing out the potential role of GluN2B inhibition in modulating the positive effects of Klotho elevation in CSDS mice. These results revealed that the upregulation of GluN2B may mediate the beneficial effects of Klotho elevation in sensitive mice [73]. It follows that the regulation of the Klotho protein by NMDA receptors may be a new strategy or molecular target for the pharmacological treatment of depression. Chen et al. (2023) showed that the effect of low-dose ketamine may be associated with the regulation of serum Klotho levels in patients with TRD [6]. In contrast, in vitro studies conducted in cultures of mouse hippocampal neurons treated for 24 h with a mixture of AMPA and NMDA antagonists (NBQX, APV—20 µM, respectively) showed that the Klotho content in neurons was significantly lower than that in vehicle-treated cells [146]. These results also confirmed that Klotho expression in neurons is modulated by glutamatergic signaling. Mazucanti et al. (2019) proposed that glutamatergic activity may regulate the release of Klotho and astrocytic aerobic glycolysis. In addition, functional (activated) AMPA and NMDA receptors are crucial for long-term potentiation (LTP), which is a synaptic transmission [146]. The molecular mechanisms underlying LTP have been widely studied regarding the physiological aspects of cognitive function, learning, and memory [147]. Evaluation of the function (manipulation) of Klotho in NMDA-dependent LTP regulation may be an additional target for elucidating the possible role of this protein in glutamatergic and synaptic transmission. Electrophysiological studies in the CA1 area of the hippocampus showed significant impairment of NMDA-dependent LTP in Klotho mutant (KO, Klotho-deficient) mice. Simultaneously, McN-A-343 (muscarinic agonist, 1.0 μg/μL in saline, i.c.v.) reversed this effect [148]. Similarly, reduced hippocampal CA1 LTP measured at 6 months of age was observed in KO (129S1/SvImJ) and overexpressing (OE, C57BL/6J) transgenic mouse models [149]. In contrast, LTP in the dentate gyrus was increased in adult (4–6 months) OE mice [17]. These data indicate that Klotho can regulate synaptic plasticity (by both pre-and postsynaptic mechanisms), but this effect is likely specific to a particular brain region [17,149]. 

The role of Klotho in the pathophysiology of depression, mainly in Glu neurotransmission, is not well understood. The promising findings presented in this section should be considered in further studies on the molecular mechanisms underlying the etiology and pharmacology of depression.

## 6. Concluding Remarks

In conclusion, the antioxidant and anti-inflammatory effects of Klotho mediated by Nrf2- NF-κB, as well as the modulation of glutamate neurotransmission function, seem to be novel signaling pathways involved in the pathogenesis of depression and cognitive impairments (see Figure 2). Importantly, the biological role of Klotho in the regulation of AMPA/NMDA receptors, EAATs expression, and their connection with oxidative–inflammatory status emphasizes the therapeutic potential of Klotho in de-repression. The hypothesis proposed in this review requires further studies (preclinical and clinical) to provide new information regarding the molecular biology of depression.

This review also highlights some limitations of the research on Klotho in nervous system diseases. Limited animal studies (particularly, animal models of depression) have demonstrated the lack of described molecular mechanisms that regulate Klotho in the central nervous system. Most human studies have focused on peripheral (serum) changes in Klotho levels in depression. The size of the group, study design, medical history, and parameters of post-mortem tissue (such as pH) can limit the relevance of funding. Importantly, defining the baseline level of Klotho in depression (independent of age) is needed.

## Figures and Tables

**Figure 1 ijms-24-15268-f001:**
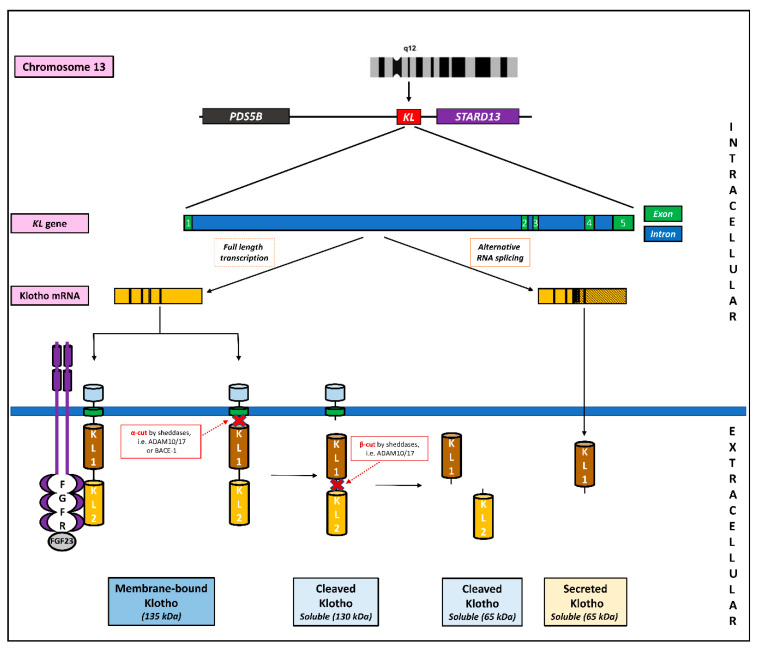
Schematic presentation of the mechanism of α-Klotho protein formation, including the location of the *KL* gene on chromosome 13 and flanking sequences PD55B and STARD13. The *KL* gene (50 Kb) is structured in five exons. The transcripts are translated into a full-length transmembrane α-Klotho consisting of three domains: cytoplasmic, transmembrane, and extracellular (two internal repeats—KL1 and KL2). The extracellular domain of α-Klotho is cleaved mainly by disintegrin and metalloproteinase (ADAM) 10 and 17 at two different sites. α-cut (also caused by β-secretases; BACE-1) and β-cut lead to the formation of the shed soluble form of Klotho (KL1-KL2, 130 kDa and KL1/1, 65 kDa, respectively). Alternative mRNA splicing leads to the formation of secreted soluble α-Klotho protein (only KL1, 65 kDa). The FGFR receptor is the primary molecular target for α-Klotho, through which it exerts its biological effect.

**Figure 2 ijms-24-15268-f002:**
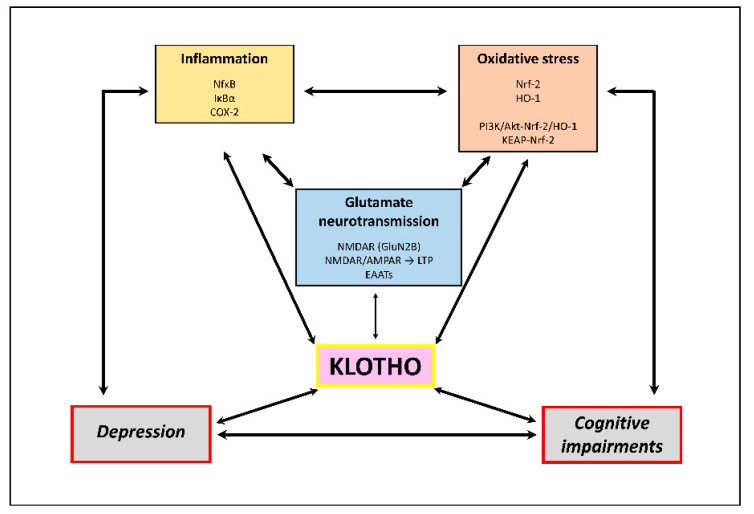
Graphical summary of potential mechanisms and links through which α-Klotho may be involved in the pathophysiology of depressive and cognitive disorders. The thickness of the lines is proportional to the strength of the evidence supporting each relationship. Only a few of the most important factors/proteins through which Klotho can exert its biological effects are presented. To improve clarity, the diagram does not show the direct relationship between depressive and cognitive disorders and glutamatergic transmission, which is well-documented.

**Table 1 ijms-24-15268-t001:** Summary of studies on the relationship between Klotho changes and depressive-like behavior using animal models of depression.

Species/Strain	Model of Depression	Samples	Methods	Results	References
Male Sprague–Dawley rats (250–300 g)	Chronic unpredictable stress (CUS; 35 days)	Choroid plexus	RT-qPCR	↓ *KL* gene in CUS rats	[72]
Male C57BL/6 J mice (8–10 weeks old)	Chronic social defeat stress (CSDS; 10 days)	Whole brain (WB), nucleus accumbens (NAc), hippocampus (Hp), prefrontal cortex (PFC)	Western Blot	↓ Klotho in WB, NAc, and Hp in mice susceptible to CSDS ↔ Klotho in PFC	[73]
Male and female Sprague–Dawley rats (10 weeks old)	Chronic unpredictable mild stress (CUMS; 3 weeks)	Hippocampus (Hp)	Western Blot fluorescence	↓ Klotho in whole Hp and CA1 pyramidal neurons in males (not in females) in the CUS group	[74]
Female Sprague–Dawley rats (10 weeks old)	Ovariectomy (OVX) + estrogen (E2) treatment (48 h or 7 days)	Hippocampus (Hp)	Western Blot immunostaining	↑ Klotho in whole Hp and CA1 pyramidal neurons after 48 h and 7 days E2 treatment	[74]

**Table 2 ijms-24-15268-t002:** The impact of genetic manipulations (*KL* gene knockdown/overexpression) on the development of depressive-like behavior and accompanying tissue changes. Summary of animal studies.

Species/Strain	Genetic Model	Results—Behavioral Tests	Samples	Results—Western Blot	References
Male CD1 mice (6 months old)	Adeno-associated virus (AAV)-mediated knockdown of *KL* in nucleus accumbens (NAc)	↓ sucrose consumption in sucrose preference test and time spent in the center in the open field test ↑ immobility time in the forced swim test and tail suspension test	Nucleus accumbens (NAc)	↓ GluN2B (total and surface, not intracellular) and PSD-95 (total) ↔ GluN1 and GluN2A (total, surface, and intracellular)	[73]
Male CD1 mice (6 months old)	Adeno-associated virus (AAV)-mediated overexpression of *KL* in NAc + chronic social defeat stress (CSDS; 10 days)	↓ interaction ratios in the social interaction test ↑ sucrose consumption in the sucrose preference test and time spent in the center in the open field test ↓ immobility time in the forced swim test and tail suspension test	Nucleus accumbens (NAc)	↑ GluN2B (total and surface, not intracellular) and PSD-95 (total) ↔ GluN1 and GluN2A (total, surface, and intracellular)	[73]
Male and female Sprague–Dawley rats (10 weeks old)	Adeno-associated virus (AAV)-mediated knockdown of *KL* in Hp + chronic unpredictable mild stress (CUMS; 2 weeks)	↓ platform area crossing and time spent in the target quadrant in the Morris water maze test (both sexes) ↓ time in center in female and total distance in male in the open field test			[74]

**Table 5 ijms-24-15268-t005:** Association between Klotho and antidepressant treatment. Summary of human studies.

Group	Samples	Methods	Results	References
MDD patients (at least 18 points on the Hamilton Depression Rating Scale (HDRS, 21 items) undergoing electroconvulsive therapy (n = 16) or a monotherapy with an antidepressant (n = 37) Healthy controls (n = 39)	Blood serum	ELISA Hamilton Depression Rating Scale (HDRS, 21 items)	No differences between the baseline Klotho levels of patients and controls, or between pre- and post-treatment in depressed patients, when treated either with electroconvulsive therapy or antidepressants	[109]
Late-life MDD patients (n = 329; ≥65 years) treated with SSRIs (esctitalopram, sertraline, paroxetine, or citalopram) for 22 weeks	Blood	DNA sequencing	↓ depressive symptoms after treatment in patients carrying at least one minor allele at rs1207568 and a worse response in patients homozygous for the minor allele at rs9536314	[111]
Patients with major depressive episode (MDD or BD, n = 8; mean age ~70 years) undergoing electroconvulsive therapy	Blood serum, cerebrospinal fluid (CSF)	ELISA	Difference between pre- and post-ECT CSF (not serum) Klotho levels CSF Klotho levels positively correlated with the number of single ECT sessions performed in each patient	[112]
Patients with major depressive episode (MDD or BD, n = 9/3; mean age 59 years) undergoing electroconvulsive therapy	Cerebrospinal fluid (CSF)	ELISA	↔ Klotho levels after ECT Klotho level not related to the Seizure Quality Index (SQI) or the ability to predict the risk of non-response (and non-remission) in ECT	[113,114]
BDI or BDII patients with moderate to severe depressive episode (n = 52) randomized to 12 bifrontal active (n = 26) or sham (n = 26) transcranial direct current stimulation (tDCS) sessions over a 6-week treatment course	Blood plasma	ELISA	↔ Klotho levels after tDCS	[115]
Patients with treatment-resistant depression (TRD) and strong suicidal ideation (n = 48) subdivided into ketamine (0.5 mg/kg; n = 24) and midazolam (0.045 mg/kg) groups	Blood serum	ELISA	↑ Klotho levels in patients on day 3 after ketamine infusion No relationship between changes in Klotho levels and changes in depressive and suicidal symptoms Higher baseline levels of Klotho were associated with a weaker antidepressant effect of low-dose ketamine during the post-infusion follow-up	[6]

## Data Availability

Data available from the corresponding author.

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
