# Peer review of "The Importance of α-Klotho in Depression and Cognitive Impairment and Its Connection to Glutamate Neurotransmission—An Up-to-Date Review"

_ijms, 2023, doi:10.3390/ijms242015268_

Round 1
Reviewer 1 Report
Dear authors,
The authors have provided a research review on the role of the Klotho protein in depression and cognitive impairment. They have also presented its molecular organization. This is a good paper; nevertheless, some points require clarification:
1- Title: The authors must decide whether they are separately discussing the role of klotho in depression and cognitive impairment or discussing depression and cognitive impairment in the context of depression.
2- Abstract: In the abstract, please briefly mention what Klotho is.
3- Introduction: I recommend that the authors make consistent use of "DD," "MDD," or "depression." These are contextually different entities and should not confuse the readers' minds. I believe it would be beneficial for the authors to consult with a clinical psychiatrist regarding this matter. Such input from a clinician also could provide valuable insights to the paper from a clinical perspective.
4- Some paragraphs are too long. For instance, the paragraphs covering pages 9-11 should be considered for division into two or three paragraphs.
5- Even though it is a non-systematic review, the authors should address their search strategy and selection criteria for the reviewed literature. This will help readers understand the methodology of their review.
6- The limitations of the current line of evidence and the limitations of this review should be mentioned.
Reviewer 2 Report
The review is quite interesting, however it is very large and not easy to read . The main question addressed by this review is the role of the Klotho protein in the pathogenesis of depressive disorders (DDs) and cognitive impairments, exploring its potential as a molecular target for future pharmacotherapy, particularly in patients with severe cognitive impairment associated with depression. The topic of the review, which explores the role of the Klotho protein in depressive disorders (DDs) and cognitive impairments, is relevant in the field of neuropsychiatry and neurobiology. It addresses the potential link between Klotho and these disorders, shedding light on a relatively recent area of research.The relevance of the topic lies in its exploration of the molecular factors that may connect depressive disorders and cognitive impairments, which are often observed together. Additionally, the review discusses the potential therapeutic implications of targeting Klotho, especially in cases of severe cognitive impairment in depression The main weakness of the review is that also it is very inclusive of past research on the topic; it does not shed light on new questions that appear after compiling, such as what are the implications of the findings..what kind of research that could be done based on their compilation. This important aspect seems to be lacking. Also, the review is large, many parts could have been reduced /summarised in the form of figures, to reduce the time needed for going through the review and make it more readable.
None.
Reviewer 3 Report
In the review article “The importance of α-Klotho in depression and cognitive impairment and its relationship with glutamate neurotransmission. Current review" presents a meta-analysis of modern data on the Klotho protein. The Review is well structured and written, it contains more than half of the references to the literature of the last 5 years, which shows the modern and relevant direction of this research. The Review presents generalizing schemes that deserve high praise. Tables 1 and 2 look of less quality. Both Tables are quite cumbersome and the authors should think about how to make them more compact, convenient and understandable to the reader. In general, the content of this Review leaves a good impression of the research of this issue and the completeness of its presentation in the format of meta-analysis analysis. Thus, after finalizing the Tables and stylistically editing the text, the Review can be published in IJMS.
The English language needs stylistic and editorial correction
Reviewer 4 Report
This review summarizes research on the role of Klotho in depressive disorders and cognitive impairment. While the review is overall interesting and informative, some issues should be addressed. In particular, there are many examples of language-related problems, which make it somewhat difficult to read it. The text would likely benefit by having it edited by a native English speaker or someone with near-native English skills. I can only name few of the problems here. There are also a few points unrelated to language that the authors should address.
1) I would eliminate “Up-to-date-review” in the title. This is self-evident.
2) Line 16: “Depressive Disorders (DD) are a neuropsychiatric diseases…”. Delete the “a”.
3) Lines 21/22: “The importance of klotho in aging processes has been best documented so far.” Replace “best” with “well”.
4) Lines 27/28: “Here, we have provided the most up-to-date research review on the role of the Klotho in DDs and cognitive impairment.” Consider eliminating this sentence, it is redundant.
5) Line 38: Replace “,i.e. 3.8%” with “,i.e., 3.8%”.
6) Lines 38/39. “DDs are the main cause of suicide and are successfully committed by approximately 700,000 people per year.” Change to “DDs are the main cause of suicide. Approximately 700,000 suicides are committed per year.”
7) Line 42: “There are numerous pharmacotherapy methods for depression, but they have many disadvantages.” Consider: “There are numerus options available for the pharmacotherapy of depression, however, their efficacy is often limited and unwanted side effects may occur.”
8) Line 49: “…ketamine with multidirectional action”. What is a “multidirectional action” ? This expression is rarely, if ever, used in the context of pharmacology.
9) Lines 63-65. “development of new drugs that reduce both depression and cognitive symptoms. Similarly, new molecular targets for this type of therapy are still under investigation being sought.” The authors should consider citing a recent review (B. Luscher et al., Trends in Pharmacological Sciences 2023;44:586-600 which focuses on such molecular targets. You could also cite a review indicating that positive allosteric modulators of alpha5 GABAA receptors are targets for treating cognitive deficits in depression (e.g., T. Prevot and E. Sibille, Molecular Psychiatry 2021;26:151-167. Moreover, “under investigation being sought” does not make sense. Consider eliminating “being sought”.
10) Line 71. At the end of the sentence there should be a reference.
11) Line 83: “numerous evidence”. Consider “considerable evidence”.
12) Line 87: “We hypothesized” should likely be “We hypothesize”.
13) Lines 92/93: ”In addition, it is presence in the brain parenchyma…”. Replace “presence” with “present”.
14) Line 108. “expressionis” should be “expression is”.
15) Line 111. “human” should be “humans”.
16) Lines 120/121. “lacks the typical for eukariotic genes TATA box” should be “lacks the TATA box, which is typical for eukaryotic genes”.
17) Line 131. Add “on” before “KL”.
18) Line 148/149. This was confirmed by a serious decrease….”. The word serious cannot be used here. Consider “significant” instead.
19) Line 197. At the end of the sentence there should be a reference.
20) Line 203. At the end of the sentence there should be a reference.
21) Lines 238/239: “which deficits are common” should be “deficits of which are common”.
22) Page 12, Table 2: “oxitocin” should be “oxytocin”.
23) Line 420. “FGF23 were elevated”. Replace “were” with “was”.
24) Lines 494/495: “This is the only hypothesis that requires in vivo investigation”. It is seomwhat difficult to understand what the authors mean. This sentence does not make sense and should be reformulated.
25) Line 510: “adnd” should be “and”.
There are quite a few language-related errors that make it more difficult or unpleasant to read the paper.
Round 2
Reviewer 4 Report
The authors did a great job responding to the previous comments. One small correction: the last word in line 50 should be changed from “receptors” to “receptor”.
Author Response
We would like to thank the Reviewer for this comment and a very detailed review of our manuscript. We are very grateful and appreciate the Reviewer's contribution to improving the quality of this review. Of course, we corrected the error in the text pointed out by the Reviewer.